# Germline Variants and Characteristic Features of Hereditary Hematological Malignancy Syndrome

**DOI:** 10.3390/ijms25010652

**Published:** 2024-01-04

**Authors:** Hironori Arai, Hirotaka Matsui, SungGi Chi, Yoshikazu Utsu, Shinichi Masuda, Nobuyuki Aotsuka, Yosuke Minami

**Affiliations:** 1Department of Hematology, National Cancer Center Hospital East, Kashiwa 277-8577, Japan; h.arai@naritasekijyuji.jp (H.A.); schi@east.ncc.go.jp (S.C.); 2Department of Hematology and Oncology, Japanese Red Cross Narita Hospital, Iidacho, Narita 286-0041, Japan; yutsu@naritasekijyuji.jp (Y.U.); smasuda@naritasekijyuji.jp (S.M.); aotsuka@naritasekijyuji.jp (N.A.); 3Department of Laboratory Medicine, National Cancer Center Hospital, Tsukiji, Chuoku 104-0045, Japan; hmatsui@ncc.go.jp; 4Department of Medical Oncology and Translational Research, Graduate School of Medical Sciences, Kumamoto University, Kumamoto 860-8665, Japan

**Keywords:** HHMS, AML, MDS, *DDX41*, *TP53*, *SAMD9*, *SAMD9L*, germline, variant

## Abstract

Due to the proliferation of genetic testing, pathogenic germline variants predisposing to hereditary hematological malignancy syndrome (HHMS) have been identified in an increasing number of genes. Consequently, the field of HHMS is gaining recognition among clinicians and scientists worldwide. Patients with germline genetic abnormalities often have poor outcomes and are candidates for allogeneic hematopoietic stem cell transplantation (HSCT). However, HSCT using blood from a related donor should be carefully considered because of the risk that the patient may inherit a pathogenic variant. At present, we now face the challenge of incorporating these advances into clinical practice for patients with myelodysplastic syndrome (MDS) or acute myeloid leukemia (AML) and optimizing the management and surveillance of patients and asymptomatic carriers, with the limitation that evidence-based guidelines are often inadequate. The 2016 revision of the WHO classification added a new section on myeloid malignant neoplasms, including MDS and AML with germline predisposition. The main syndromes can be classified into three groups. Those without pre-existing disease or organ dysfunction; *DDX41*, *TP53*, *CEBPA*, those with pre-existing platelet disorders; *ANKRD26*, *ETV6*, *RUNX1*, and those with other organ dysfunctions; *SAMD9*/*SAMD9L*, *GATA2*, and inherited bone marrow failure syndromes. In this review, we will outline the role of the genes involved in HHMS in order to clarify our understanding of HHMS.

## 1. Introduction

Most hematologic malignancies are thought to spontaneously arise due to acquired genetic lesions in hematopoietic stem and precursor cells (HSPCs) [1]. However, in some cases of acute myeloid leukemia (AML) and myelodysplastic syndrome (MDS), a hereditary (mainly autosomal dominant) predisposition has been observed [2,3]. Typically, a family in which two or more first- or second-degree relatives have developed acute leukemia (AL), myeloid malignancies, characteristic cytopenias, or either MDS or AML, is defined as “familial MDS/AML”, or, more broadly, hereditary hematologic malignancy syndrome (HHMS) [4,5,6]. The field of HHMS has gained increasing recognition among clinicians and scientists worldwide. Both myeloid and lymphoid malignancies may be present in individuals or families with these syndromes. Genetic predisposition should be considered in patients who present with bone marrow failure, MDS, or AML at a young age or who present with unexpected hematologic toxicity during treatment for malignancy at a young age [7,8]. Identifying characteristics of such patients include physical abnormalities, endocrine abnormalities, short stature, stunted growth, and immunodeficiency in patients with hematologic abnormalities such as cytopenia, unexplained macro-erythroblastosis, or overt malignancy. A genetic MDS/AML predisposition may also be indicated by a family history of first- or second-degree relatives with malignancy, cytopenia, congenital abnormalities, or excessive toxicity from chemotherapy or radiation therapy [9]. However, the absence of characteristic clinical features or a negative family history does not exclude the presence of a germline MDS/AML syndrome. Germline variants may occur de novo or result from parental gonadal mosaicism [10]. HHMS often shows marked inter- and intra-familial differences in latency, phenotype, expression, and penetrance. For example, some germline MDS syndromes lack obvious syndromic features or have variable penetrance or delayed expression. Cytogenetic clonal abnormalities common to certain inherited MDS disorders may warrant further investigation [11]. MDS with monosomy 7 frequently occurs in patients with germline variants in GATA-binding factor 2 (*GATA2*), sterile alpha motif domain containing 9 (*SAMD9*), sterile alpha motif domain containing 9 like (*SAMD9L*), or hereditary bone marrow failure syndrome [12]. Moreover, the involvement of hematopoietic transcription factor genes, such as CCAAT enhancer binding protein alpha (*CEBPA*), *GATA2*, runt-related transcription factor 1 (*RUNX1*), ankyrin repeat domain containing 26 (*ANKRD26*), and ETS variant transcription factor 6 (*ETV6*), is traditionally associated with solid tumors such as MutS homolog 6 (MSH6) and breast cancer gene 1 (*BRCA1*). Moreover, the recently identified genes DEAD-box helicase 41 (*DDX41*), *SAMD9*, *SAMD9L* are involved in leukemogenesis [13,14,15]. Many are found to be non-symptomatic and occur in various age groups. Studies suggest that about 10% of children and adults with MDS or AML may have heritable variants [5]. Importantly, these germline genetic abnormalities are not exclusive to the patient and may be shared by blood relatives, necessitating the screening of blood relatives. As our diagnostic capabilities in HHMS improve, we now face the challenge of incorporating these advances into clinical practice with MDS/AML patients and learning how to optimize the management and surveillance of patients and asymptomatic carriers [16].

The discovery of novel syndromes combined with the clinical, genetic, and epigenetic profiling of tumor samples has highlighted unique patterns of disease progression in HHMS. Despite these advances, causative lesions are identified in fewer than half of familial cases, and evidence-based guidelines are often inadequate. In the 2016 revision of the WHO classification, a new section was added for myeloid neoplasms with a germline predisposition, including cases of MDS, myeloproliferative neoplasms (MPN), and ALs that develop on a background of predisposing germline variants [17]. As part of the diagnosis, specific underlying genetic abnormalities or predisposing syndromes should be considered. The major syndromes can be categorized into the following three groups: those without preexisting disease or organ dysfunction [e.g., *DDX41*, tumor protein p53 (*TP53*), and *CEBPA*], those with pre-existing platelet disorders [e.g., *ANKRD26*, *ETV6*, and *RUNX1*], and those with organ dysfunction [e.g., *SAMD9*/*SAMD9L*, *GATA2*, and inherited bone marrow failure syndromes (IBMFSs)]. This review will outline the genes involved in the above HHMS (Table 1).

## 2. Myeloid Neoplasms without a Preexisting Disorder or Organ Dysfunction

### 2.1. Myeloid Neoplasms with a Germline DDX41 Variant

RNA helicases are a series of enzymes that remodel RNA–RNA or RNA–protein interactions in an NTP-dependent manner. Humans have more than 70 helicases that are classified into superfamily (SF) 1 and SF2 based on differences in sequence motifs within the helicase core domain [59,60]. SF1 includes Upf1-like RNA helicases, while SF2 includes the DEAD-box, DEAH-box/RNA helicase A-like, Ski2-like, and RIG-I-like families, with the DEAD-box family RNA helicases being the most numerous. While the DEAH-box RNA helicases are thought to translocate along the substrate RNA for remodeling, DEAD-box RNA helicases unwind substrate RNA locally; the mechanism of action of each is thus different, but they both play roles in virtually all processes that require RNA conformational changes, such as RNA transport, translation, RNA degradation, RNA splicing, and ribosome synthesis. As a single RNA helicase often exerts enzymatic activity in multiple cellular processes, it remains difficult to fully elucidate the pathogenesis of diseases due to abnormalities in RNA helicases.

In myeloid neoplasms, pathogenic variants in the gene encoding DDX41, a DEAD-box RNA helicase, are found in about 5% of cases [61]. It was recently shown that up to 13% of myeloid neoplasms have a genetic background [62], of which *DDX41* variants account for about 80% of cases. MDS and AML occur in individuals with a heterozygous germline frameshift variant or a missense variant within the DEAD-box domain of *DDX41* by later acquiring a somatic variant in the other allele, typically p.R525H (or p.G530D, etc., in a few cases) within the helicase domain [61,63,64] (Figure 1A). While many myeloid neoplasms with a genetic background develop at younger ages than those without a known genetic background, myeloid neoplasms with *DDX41* variants are characterized by a late disease onset (mean age, 65 years) [22,23], which may hinder the identification of this gene as one of the genes responsible for genetic predisposition for myeloid leukemogenesis. In addition, the disease with a *DDX41* variant is characterized by male dominancy, fewer proliferating tumor cells, hypoplastic bone marrow, and unique co-existing gene mutational patterns as compared to those in other myeloid neoplasms [65,66], with only *DDX41* variants being identified in many cases [61], suggesting a unique disease pathogenesis of myeloid neoplasms with *DDX41* variants. In contrast, the disease phenotype may differ between cases with a single *DDX41* variant and biallelic variants [67], and a report suggest that there is no clear difference in disease phenotype between cases with known pathogenic *DDX41* variants and variants of unknown significance (VUS) [68]. Consequently, it is necessary to establish a validation system and database that can accurately interpret the significance of individual variants.

A combination of germline and somatic *DDX41* variants confers myeloid disease development.

Hematopoietic cells with a germline *DDX41* variant acquire a somatic *DDX41* variant at an advanced age. Myeloid neoplasms are thought to develop shortly after biallelic *DDX41* variant acquisition, with or without the addition of a limited number of somatic variants in DNA repair-related genes, including *CUX1* and *TP53*. It is also suggested that minor clones with biallelic *DDX41* variants affect hematopoiesis by interfering with other cells [37].

B.R-loop formation and its consequence.

R-loop accumulation due to impaired RNA splicing or other causes increases DNA replication stress and innate immune response, resulting in deficient hematopoiesis and leukemogenesis.

The prognosis of myeloid neoplasms with *DDX41* variants is not necessarily worse than for those without a known genetic background, regardless of the tendency to be categorized as high-risk. However, the development of disease at advanced ages often makes intensive treatment difficult. Several cases of donor-derived secondary leukemia in patients who received allogeneic hematopoietic stem cell transplantation (HSCT) have been reported [18,69,70,71]; thus, treatment decisions require the careful consideration of genetic background. Recent reports describe the development of acute lymphocytic leukemia and solid cancers in individuals with *DDX41* variants [72,73], but the extent to which *DDX41* variants are involved in such diseases remains controversial [64].

*DDX41* has been shown to be essential for hematopoiesis, with homozygous *Ddx41* knockout mice being embryonic lethal, although heterozygous mice show no remarkable abnormalities [74,75]. Several mechanisms have been proposed for the actions of *DDX41* variants in the development of myeloid neoplasms. It has been reported that R-loop, a nucleic acid structure on the genome consisting of a DNA/RNA hybrid and single-strand DNA, aberrantly accumulates in MDS with RNA splicing abnormalities, regardless of the type of responsible gene [76,77,78,79], and that R-loop accumulation causes DNA replication stress, DNA damage, and abnormal mitosis. Recently, *DDX41* has also been shown to be involved in R-loop regulation [80,81,82], and it is suggested that R-loop accumulation due to dysfunction or decreased expression of *DDX41* is involved in impaired hematopoiesis and aberrant innate immune responses (Figure 1B). One of the major functions of *DDX41* is RNA splicing [19]. However, considering that *DDX41* variants develop de novo AML in addition to MDS, *DDX41* is thought to play different roles from those of typical RNA splicing factors associated with MDS development. Indeed, while SRSF2, SF3B1, and U2AF1 are all involved in the recognition of pre-mRNA 3′ splice sites with U2 snRNP [83], *DDX41* has been shown to be incorporated into the spliceosome at the C complex stage, a late complex of the activated spliceosome [82,84]. Regarding the relationship between *DDX41* and R-loops, there are reports showing that *DDX41* can unwind R-loops on its own [81,85], while it has also been suggested that impaired *DDX41* function leads to reduced efficiency of RNA splicing, thus resulting in conditions that facilitate R-loop formation [82]. The accumulation of R-loop has been shown to give rise to an excessive innate immune reaction mediated through the cGAS-STING signaling pathway, consequently inducing increased hematopoietic stem/progenitor cells [80]. However, the mechanisms by which R-loops activate the cGAS-STING pathway remain inconclusive. Recently, it was reported that DNA/RNA hybrids derived from R-loops are transported to the cytoplasm and thus trigger an innate immune response [86]. The relevance of this observation to impaired hematopoiesis caused by *DDX41* variants is of interest.

*DDX41* is also reported to promote the processing of small nucleolar RNA (snoRNA) from introns [75]. Some snoRNA are coded within introns of ribosomal protein genes and mature after being processed from the introns [87,88]. snoRNAs are classified into boxC/D type and boxH/ACA types depending on their sequences; the former catalyzes 2′-O-methylation and the latter is responsible for catalyzing the pseudouridylation of uridine residues in ribosomal RNA, thereby promoting ribosomal biogenesis. Thus, loss of function (LOF) or expression of *DDX41* impairs ribosomal biogenesis [66,89]. Although the involvement of DDX41 in ribosomal biogenesis has been reported by other research groups, the process involving DDX41 may be different from processes involving snoRNA processing.

Recently, myeloid neoplasms with germline *DDX41* variants were shown to have a higher proportion of somatic *CUX1* variants compared with those without a known germline background [61]. CUX1 is a transcription factor [90] that has also been shown to be directly involved in DNA damage repair by recruiting histone-modifying enzymes to damaged DNA regions [91]. Given that cells lacking sufficient CUX1 function can enter mitosis without completing DNA damage repair, the likelihood that the loss of *DDX41* function or expression causes DNA replication stress is further increased. However, further studies are clearly needed to fully elucidate the mechanisms by which *DDX41* variants lead to myeloid neoplasms.

### 2.2. Li-Fraumeni Syndrome (LFS)

*TP53* is one of the most frequently mutated genes, especially in adult-onset cancers. Genome sequencing of various human cancer cells has revealed that 42% of cases carry *TP53* variants [92]. The p53 protein is a transcription factor that can activate the expression of multiple target genes, plays an important role in the regulation of the cell cycle, apoptosis, and genomic stability, and is widely known as “the guardian of the genome”(Figure 2) [93,94]. The evidence accumulated to date suggests that p53 also regulates cell metabolism, ferroptosis, tumor microenvironment, and autophagy, which each contribute to tumor suppression [94]. Genomic instability caused by deletions and variants in *TP53* may lead to accumulated gene mutations, causing gain of function (GOF) in the oncogene and LOF in the tumor suppressor gene [95]. p53 variants confer metabolic plasticity to cancer cells, promoting adaptation to metabolic stress and increasing the possibility of proliferation and metastasis [96].

The major type of *TP53* variant is a missense variant producing a single amino acid substitution, with the DNA-binding domain (DBD) being the most mutated region [97]. Structural variants can reduce the thermostability of the protein, resulting in protein misfolding at physiological temperatures and a loss of its ability to bind DNA [98]. These variants not only bind wild-type p53 and cause dominant-negative (DN) effects, but may also be converted to oncogenic proteins via GOF, promoting various cellular responses such as carcinogenesis, cancer cell proliferation, invasion, metastasis, tumor microenvironment establishment, genomic instability, and metabolic reprogramming [99,100]. p53 is mutated and inactivated in most malignancies, making it a very attractive target for the development of new anti-cancer drugs [101]. Until recently, however, p53 was considered an undruggable target, and the progress made in p53-targeted therapeutics has been limited.

LFS is caused by a germline variant in the *TP53* gene and is characterized by an increased risk of developing various solid tumors and hematologic malignancies at a young age [102,103]. LFS affects all ethnicities and has an estimated incidence of 1:5000 [27]. LFS is inherited in an autosomal dominant manner, although de novo inheritances occur in 7–20% of cases. Nearly 100% of individuals develop cancer by the age of 70, with the median age of first cancer at 20 to 30 years [26]. The tumor spectrum includes soft-tissue sarcomas, premenopausal breast cancer, central nervous system tumors, adrenocortical carcinomas, and pancreatic tumors, as well as MDS and lymphoid and myeloid malignancies. Germline *TP53* variants are found in approximately 50% of pediatric patients with hypoploid acute lymphoblastic leukemia (ALL) and are associated with poor outcomes [104,105]. In the Le-Fraumeni lineage, leukemia is relatively uncommon, with only approximately 4% of children and adolescents presenting with hypodiploid ALL, treatment-related, or de novo MDS/AML [29].

As causal therapy is not available, the primary focus for improving the prognosis is early cancer detection. To this end, current cancer surveillance recommendations include a series of examinations including regular imaging beginning at birth [102]. As radiation exposure may lead to an increased (secondary) tumor risk, computed tomography and X-ray examinations should be avoided for as long as possible. Because annual whole-body magnetic resonance imaging has no radiation exposure and yet a high sensitivity for many tumors, it forms the basis of the recommended imaging [102].

### 2.3. AML with a Germline CEBPA Variant

The *CEBPA* gene is located on chromosome 19q13.1 and gene variants are a common genetic alteration in AML. Patients present with de novo AML [French American-British (FAB) classification; AML M1, M2, and M4 subtypes] and a group of differentiation abnormalities [106].

The single-exon gene *CEBPA* encodes CEBPa, which is the founder of the 6-CEBP family of transcription factors (TFs) [107]. All CEBP TFs contain a basic leucine zipper (bZIP) domain at the C terminus and form a subgroup within the leucine zipper family of TFs [108]. The CEBPa zipper domain is required for dimerization, and the adjacent basic region is responsible for DNA binding, thereby promoting the transcription of target genes [109]. The N terminus is unique to CEBPa, containing two transactivation domains that regulate transcription control and protein interaction [109]. CEBPa generates two isoforms from alternative initiation codons: the long isoform (p42) is 358 aa, and the short isoform (p30) is 239 aa and lacks a transactivation domain [110]. The p30 isoform maintains dimerization and DNA binding capacities, and, hence, can inhibit p42 activity. Both isoforms are coexpressed in a range of tissues, with p42 generally being more abundant [110]. Germline and somatic variants in *CEBPA* are clustered at the N terminus or within the C-terminal bZIP domain. These germline variants are generally frameshift or nonsense variants near the amino terminus of the encoded protein. Somatic variants in *CEBPA* often occur in the other allele, leading to a biallelic variant in *CEBPA*. This triggers the development of AML [31]. Commonly, the germline variant affects the N terminus, whereas the acquired variant arises in the C-terminal bZIP region [108]. The *CEBPA* variants that predispose to AML are generally considered to have a dominant-negative effect. The N-terminal truncating variants destroy p42, and the C-terminal variants abolish DNA binding or dimerization [108].

*CEBPA*-associated familial AML is defined as the presence of heterozygous germline *CEBPA* pathogenic variants in AML patients and/or in families with one or more AML patients. In contrast, sporadic *CEBPA*-associated AML is defined as AML in which the *CEBPA* pathogenic variant is identified in leukemic cells and not in non-leukemic cells [111]. AML with germline *CEBPA* variants generally occurs in autosomal-dominant inheritance without preceding abnormal blood cell counts or myelodysplasia [112]. Approximately 10% of *CEBPA*-associated AMLs have been shown to carry germline *CEBPA* variants [2]. In contrast to the incomplete penetrance observed in other HHMSs, germline *CEBPA* variants cause AML with almost complete penetrance (lifetime risk estimated to be >80%) [113]. Less than 20 families have been reported to have germline *CEBPA* variants [32]. In the majority of *CEBPA*-associated familial AML, the age of onset appears to be earlier than in sporadic *CEBPA*-associated AML [111]. Onset usually occurs in the 20th or 30th year of life, and many patients develop AML before 50 years of age; the median age of onset for AML is 24.5 years [34]. The prognosis of *CEBPA*-associated familial AML appears to be better than that of sporadic *CEBPA*-associated AML [114,115]. Patients with *CEBPA*-associated familial AML with a cured initial presentation are at high risk of developing additional independent leukemic episodes in addition to the risk of relapse from a pre-existing clone; the clinical observation that AML patients with *CEBPA* variants are more likely to develop a secondary leukemia despite their favorable prognosis is likely due to this pattern of progression [37]. Lifelong surveillance is recommended in patients with familial AML because of the high risk of late leukemia relapse [16]. It is important to avoid the use of allogeneic or consanguineous donors for HSCT without prior evaluation of the donor’s germline *CEBPA* pathogenic variant [116].

### 2.4. Myeloid Neoplasms with Other Germline Variants (ATM and CHEK2)

Deficiencies in the homologous recombination (HR) pathway can lead to defective DNA damage responses, and this can occur through inherited germline variants in HR pathway genes, such as checkpoint kinase 2 gene (CHEK2) and the ataxia telangiectasia mutated gene (ATM). The proper repair of DNA double-strand breaks (DSBs) is a core element of the maintenance of genomic stability, directed through three pathways active in most human cells: (1) homologous recombination (HR); (2) canonical non-homologous end joining (NHEJ); and (3) alternative NHEJ [117]. Canonical NHEJ is the simplest DNA repair mechanism. It involves directly adjoining DSBs through the binding of the Ku-80-Ku7p proteins to the fragmented DNA ends, followed by the recruitment of DNA-dependent protein kinases, which then activate ligase IV and co-factors which seal the DNA break. The alternative NHEJ mechanism involves the recruitment of PARP to the DNA ends, ending in the DNA DSB being sealed by Ligase I and III [118]. Although NHEJ effectively repairs DNA DSBs, it does not involve the usage of a complementary DNA template, and, as such, is error-prone, inducing chromosomal abnormalities and chromothripsis [118]. In contrast, HR is the most error-free of the DNA repair pathways, since it uses a complementary DNA template available during S-phase to correct the detected DNA lesion [119]. The HR pathway is engaged when the MRE11-RAD50-NBS1 protein complex is recruited to the fragmented DNA ends, which subsequently recruits ATM serine/threonine kinase. Activated ATM then phosphorylates the checkpoint kinase 2 (CHK2) protein, resulting in the downstream activation of a series of proteins, including CDC25C, p53, BRCA1/2, and cyclin-D kinases, which coordinate template-based DSB repair, cell-cycle arrest, and potentially apoptosis [117,118]. Another important feature of the DSB response is the induction of cell cycle checkpoint arrest, mediated during the S or G2 phase by ATR serine/threonine kinase and ATM, and it is essential for allowing the cell to re-enter mitosis after successful DSB repair [118].

Germline pathogenic and likely pathogenic gene variants that result in loss of function (LOF), such as ATM and CHEK2 variants, have been identified and characterized at several levels of the HR pathway. The broad consequence of a variant in one of these genes is a defective HR pathway, with consequent reliance on error-prone NHEJ mechanisms for DNA repair. The downstream result of using error-prone DNA repair pathways is an accumulation of somatic chromosomal abnormalities and DNA changes, particularly within rapidly dividing cells (e.g., epithelial, mammary, and hematopoietic), with an increased risk for the development of overt malignancy. Germline mutations in these genes have been well characterized as risk factors for breast [120], prostate [121,122], and pancreatic [123] cancers.

CHK2 protein is essential to the transmission of the DSB signal from ATM to downstream effectors CDC25C, p53, BRCA1/2, cyclin-D kinases, and others via phosphorylation. A variety of mutation types in CHEK2 have been identified, including splice site, missense, and frameshift, without a predisposition towards mutational hotspots [124]. Although the majority of patients carrying CHEK2 variants are in the heterozygous state, individuals with homozygous LOF CHEK2 do occur and have a Li-Fraumeni like phenotype. Heterozygous LOF CHEK2 variants are moderate penetrance risk factors for solid organ malignancies, including breast [125], prostate [126], renal cell carcinoma [127], papillary thyroid cancer [128], colorectal cancer [129], and pancreatic cancer [130]. LOF CHEK2 variants are increasingly recognized as risk factors for myeloid malignancies, including MPNs [131,132,133], MDS [134,135], and AML [136,137,138]. Germline CHEK2 variants have also been identified as risk factors for lymphoid malignancies [139].

Germline LOF mutations in the ATM gene have long been associated with early-onset myeloid malignancies, in addition to solid tumors such as breast and pancreatic cancers [140]. Loss of ATM function generates a greater risk of chromosomal translocations and other deleterious mutations associated with myeloid leukemia development [141]. Patients carrying LOF ATM variants in the homozygous or compound heterozygous states present with Ataxia Telangiectasia (A-T), an autosomal recessive disorder characterized by a 50- to 150-fold increased risk of cancer development, and also cerebellar degeneration, telangiectasia, immunodeficiency, and radiation sensitivity [142]. However, most individuals with germline deleterious ATM variants are heterozygous carriers with a 2- to 13-fold increased risk for early-onset cancer development but do not have other features of A-T [143]. The role for ATM mutations in myeloid malignancies remains in evolution and is less well characterized than for CHEK2. However, pathogenic ATM variants have been identified at diagnosis in several patients with de novo AML [62]. Intact ATM function has been well established as being critical for hematopoietic stem cell function [144], and ATM function and the associated signaling axis have been shown in vitro to modulate pathogenesis in AML [145,146]. In contrast to myeloid malignancies, germline ATM variants, either in the heterozygous state or in the context of true A-T with biallelic ATM variants, have been strongly associated with the development of lymphoid malignancies [147,148,149].

## 3. Myeloid Neoplasms with Preexisting Platelet Disorders

Most predisposition syndromes are associated with specific hematopoietic cell lineage abnormalities, and each exhibits a different tumor profile. For example, germline variants in *RUNX1*, *ANKRD26*, and *ETV6* all predispose to thrombocytopenia and hematologic malignancies [150]. However, there are marked differences in cancer predisposition: the *ANKRD26* variant predisposes to myeloid malignancies, *ETV6* predominantly predisposes to B-cell ALL, and *RUNX1* is associated with myeloid malignancies, and, to a lesser extent, predisposes to T-cell ALL [151]. Three different types of germ cell lineage predisposition are associated with highly variable penetrance in both myeloid and lymphoid systems. In both myeloid and lymphoid leukemias, the disease phenotype is likely influenced by both intrinsic and extrinsic cellular factors [150].

### 3.1. Myeloid and Lymphoid Neoplasms with a Germline RUNX1 Variant

*RUNX1* encodes a heterodimeric transcription factor essential for hematopoiesis, megakaryopoiesis, and platelet function [152]. It functions as a transcriptional activator for some genes and a transcriptional repressor for others. Somatic variants in *RUNX1* are among the most common variants in adults and children with ALL, AML, or MDS, including recurrent fusions in B-ALL (*ETV6*-*RUNX1*) and AML (*RUNX1*-*RUNX1T1*) [41]. *RUNX1* was identified as a gene located at a truncation site on chromosome 21 in t (8;21), which is found in AML [153]. Somatic variants in the *RUNX1* gene are one of the most frequently identified variants and have been identified in patients with various myeloid malignancies, including MDS, MPN, and AML [40]. In most cases, these *RUNX1* variants are considered “subclonal variants” [154]. A high frequency of *RUNX1* variants (30–50%) has been reported in treatment-related and radiation-related MDS and AML [155,156]. It is generally believed that *RUNX1* variants lead to a loss of *RUNX1* function [157]. In contrast, germline variants in the *RUNX1* gene cause familial myeloid malignant platelet disorders (FPD/AML) with autosomal dominant inheritance, typically presenting with quantitative/qualitative platelet defects and a predisposition to myeloid malignancies like MDS and AML [158]. In this case, heterozygous inherited *RUNX1* variants play a fundamental role in the etiology of FPD/AML [159]. However, these inherited *RUNX1* variants are not sufficient to cause leukemia. It is thought that the accumulation of various variants, such as the CDC25C biallelic *RUNX1* variant, and the *TET2* variant, causes progression to preleukemic clones and eventually leads to the development of hematologic malignancies [38,160].

Germline variants in *RUNX1* are among the most frequently detected variants in the pathogenesis of HHMS [38]; the *RUNX1* gene encodes a DNA-binding subunit that contains a highly conserved runt-homology domain (RHD) for sequence-specific DNA binding [161]. Truncation lesions occur throughout the gene, but missense variants within the RHD are the most common. Others include nonsense, frameshifts, duplications, partial or total gene deletions, and gene rearrangements. Many *RUNX1* variants cause haploinsufficiency [157]. *RUNX1* variants cause defects in hematopoietic differentiation, resulting in decreased hematopoietic progenitor cell numbers and abnormal megakaryocyte differentiation. Tumorigenesis is most commonly caused by the somatic second hit of *RUNX1*. Typical clinical features of FPD/AML are gradual thrombocytopenia, aspirin-like qualitative platelet abnormalities, and a tendency to develop hematologic tumors [162].

Approximately 20–60% of FPD/AML families develop hematologic neoplasms during their lifetime [162]. More than 250 families have been reported to have germline RUNX1 variants. The latency period to transformation is relatively long, with the average age at diagnosis reported to be 33 years (maximum 76 years) [41]. Similar to what is observed in sporadic hematologic malignancies, additional acquired genetic events cooperate with the hereditary *RUNX1* variant to progress the manifestation of the malignant phase. A comparative international cohort of germline RUNX1 variant carriers without and with hematological malignancies (HM) identified striking heterogeneity in rates of early-onset clonal hematopoiesis (CH), with a high prevalence of CH in RUNX1 carriers who did not have malignancies (carriers without HM). In RUNX1 carriers without HM with CH, TET2, PHF6, and BCOR were reported to be recurrently mutated in RUNX1-driven malignancies, suggesting that CH is a direct precursor to malignancy in RUNX1-driven HHMS [163].

Although most cases develop MDS or AML, other phenotypes have also been reported, including secondary leukemia, T-cell acute lymphoblastic leukemia (T-ALL), and non-Hodgkin lymphoma (NHL) [162]. Interestingly, the location of variants within the *RUNX1* gene does not seem to affect disease phenotype among individuals, and phenotypic heterogeneity is often observed even within families with lesions of the same germ lineage [93].

### 3.2. Myeloid Neoplasms with a Germline ANKRD26 Variant

*ANKRD26* is a gene located at 10p12.1 that regulates megakaryocyte development and thrombocytopenia [164]. RUNX1 and FLi1 co-regulate ANKRD26 by binding to the *ANKRD26* promoter and repressing gene activity [165]. *ANKRD26*-related thrombocytopenia (*ANKRD26* RT) is an autosomal dominant thrombocytopenia caused by a single nucleotide substitution in the *ANKRD26* gene, characterized by quantitative and qualitative platelet disorders and an increased risk of MDS and AML [166]. *ANKRD26* encodes a protein with an ankyrin repeat domain at its N-terminus and is thought to function in protein–protein interactions; while the function of the ANKRD26 protein is unknown, expression profiling has demonstrated its presence in megakaryocytes [166]. Germline variants in *ANRK26* are usually point mutations located in the 5′ untranslated region (UTR) of the gene, although deletions and point mutations within the coding region have also been reported [167]. Variants in the 5′UTR affect the binding of repressive transcription factors such as RUNX1 and FLi1 to this regulatory region, abnormally increasing the expression of *ANKRD26* and impairing platelet production [150]. The age of diagnosis generally ranges from early 20s to 70s. The incidence of myeloid malignancies is high in these patients, with an estimated 5% for AML, 2.2% for MDS, and 1.3% for chronic myeloid leukemia, with an estimated risk of these malignancies of 23, 12, and 21 times that of the general population, respectively [14].

### 3.3. Myeloid and Lymphoid Neoplasms with a Germline ETV6 Variant

Patients with thrombocytopenia 5, an autosomal dominant disorder of thrombocytopenia with bleeding tendency, usually present in childhood and have been found to have germline variants in *ETV6* [168]. Clinical features include thrombocytopenia, abnormal platelet function, and increased bleeding tendency [49]. Leukemia is estimated to occur in about 30% of carriers, most commonly in ALL, but more than 30 translocation partners of *ETV6* have been reported in AML, MDS, MPN, and T-cell lymphomas. *ETV6* is one of the most commonly translocated genes in human AL and MDS [169]. ALL is more frequent, especially in B-ALL (0.8% of unselected childhood B-cell ALL). The ratio of lymphoid versus myeloid malignancies is roughly 2:1. Age ranges from 8 to 82 years and it seems to occur at a younger age than usual but is not yet defined [45,49,50].

*ETV6* is located on chromosome 12p13.2 and encodes a transcriptional repressor important for hematopoiesis, megakaryopoiesis, and embryogenesis, and it is involved in angiogenesis, cell growth, and differentiation [170]. The gene encodes an N-terminal or C-terminal zinc finger, but the majority of variants are clustered within the DNA-binding ETS domain. Somatic rearrangements (most commonly with *RUNX1*), deletions, and sequence variants are observed in ALL. Second-hit variants (especially deletions) in *ETV6* are common in *ETV6-RUNX1* rearranged leukemias [171]. In addition, somatic rearrangements with *RUNX1* are observed in a quarter of ALL patients [172]. Studies using umbilical cord blood from healthy newborns have shown that *ETV6-RUNX1* translocations can occur in more than 1% of the healthy population [173].

## 4. Myeloid Neoplasms with Other Organ Dysfunction

### 4.1. Myeloid Neoplasms with a Germline SAMD9/SAMD9L Variant

*SAMD9* and *SAMD9L* are a homologous gene pair at the head and tail of 7q21 and are interferon-inducible genes that are widely expressed in human tissues [55,174]. Both negatively regulate cell proliferation and function as tumor suppressors. Genetic variants in *SAMD9/SAMD9L* were initially shown to cause multisystem syndromes characterized by various neurological and/or endocrine abnormalities, as well as MDS with monosomy 7 and del7q [55,175]. Little is known about the biochemical activity of the SAMD9 and SAMD9L proteins and their domain structures, but they cluster in the latter half of the protein, in or near the putative P-loop [176]. The SAMD9 and SAMD9L proteins appear to be involved in endocytosis and cytokine signaling [177,178]; moreover, they have been reported to play a role in antiviral responses, similar to *DDX41*. Specifically, *SAMD9* and *SAMD9L* are known to be host-restricted factors in poxvirus infection [179,180].

Germline variants in these genes are strongly associated with monogenic and familial pediatric MDS and potential full or partial deletions of adult chromosome 7(Figure 3) [53]. Germline variants in *SAMD9* or *SAMD9L* are heterozygous gain-of-function missense variants, leading to proliferative arrest when expressed exogenously in the cell [174]. Carriers are at high risk for MDS and AML with cytopenia and monosomy 7/del7q. Many other patients who do not develop monosomy 7 acquire somatic variants in *SAMD9* or *SAMD9L* resulting in the loss of function of the mutant protein [181]. The overexpression of *SAMD9* or *SAMD9L* results in decreased proliferation and increased apoptosis, ultimately leading to the hypocellular phenotype being observed in patients. The effects on ribosome biology, DNA damage, and the resulting genomic instability are thought to promote the observed apoptotic phenotype [182,183] and ultimately lead to reduced bone marrow cellularity. Unrepaired DNA defects in hematopoietic cells cause significant long-term functional disruption and are a major driving force for the accumulation of further variants, thus promoting clonal expansion and malignant transformation [184,185,186].

Germline variants in *SAMD9* cause a syndrome represented by the acronym MIRAGE; MIRAGE syndrome is an autosomal-dominant multisystem disorder characterized by six core features [187,188,189,190,191]. The features include bone marrow failure, progression to MDS and AML, infection, intrauterine dysplasia, adrenal hypoplasia, genital abnormalities, and enteropathy (chronic diarrhea with colonic dilatation). Germline variants in *SAMD9L* cause ataxia-pancytopenia syndrome, an autosomal dominant disorder with early onset gait and balance disturbances, nystagmus, mild pyramidal signs, and marked cerebellar atrophy [192,193,194,195]. Hematologic abnormalities include pancytopenia, bone marrow failure, and progression to MDS and AML. Germline variants in these two genes are found in 8–17% of pediatric MDS cases and more than 110 individuals have been reported to carry these germline variants [55]. It occurs mainly in childhood, but the average age of onset is not yet defined.

### 4.2. Myeloid Neoplasms with a Germline GATA2 Variant

Hematological malignancies affecting either the lymphoid or the myeloid lineages involve epigenetic mutations or dysregulation in the majority of cases. These epigenetic abnormalities can affect regulatory elements in the genome, and, particularly, enhancers. Recently, large regulatory elements known as super-enhancers (SE), initially identified for their critical roles in the cell-type specific expression regulation of genes controlling cell identity, have been shown to also be involved in tumorigenesis in many cancer types and hematological malignancies via the regulation of numerous oncogenes. Enhancer and SE hijacking refers to a mechanism by which an abnormally overexpressed TF binds to an inactive or poised enhancer already located near a given oncogene, recruiting other factors and chromatin remodelers. This binding allows the aberrant activation of the considered enhancer/SE, and, thereby, upregulates its associated oncogene. An example of such enhancer hijacking is provided by AML with the GATA2 SE translocated near the *EVI1* promoter. A single enhancer contained within this GATA2 SE is composed of MYB binding sites, strongly required for *EVI1* overexpression in AML cells. In addition, the mutation of this MYB binding site within this specific SE leads to myeloid differentiation, as well as cell death [196,197].

*GATA2* is a zinc finger transcription factor that plays important roles in hematopoiesis, the homeostasis of hematopoietic stem cells (HSC), and lymphocyte development, specifically interacting with *RUNX1* to control HSC survival [198]. *GATA2* haploinsufficiency is caused by a missense variant or deletion in the *GATA2* located on chromosome 3q21.3 [199]. Other causative variants have been detected throughout the gene, including nonsense, frameshift, splice site, and synonymous variants that cause splice abnormalities, as well as variants that target enhancers deep within introns [200]. *GATA2* haploinsufficiency is an autosomal dominant inherited bone marrow failure and immunodeficiency syndrome predisposing to MDS and AML. The syndrome results from loss-of-function variants or deletions in the *GATA2* gene [201]. Notably, *GATA2* deficiency syndromes (G2DSs) show marked heterogeneity in inter- and intra-familial phenotypes, all within the spectrum of the single condition G2DS [13,202].

Phenotypes range from isolated chronic neutropenia to MDS/AML, bone marrow failure, severe immunodeficiency, and alveolar proteinosis. Patients may present with isolated neutropenia and bone marrow failure without syndromic features or family history [203]. Atypical mycobacterial infections, viral, and fungal infections are common, often overlapping with prolonged neutropenia, monocytopenia, B-cell deficiency, NK-cell deficiency, monocytopenia with Mycobacterium avium complex (MonoMAC) syndrome, or dendritic cell-monocyte-B-NK lymphocyte (DCML) deficiency [204,205]. Other symptoms include sensorineural hearing loss and lymphoedema (Emberger syndrome) [206,207].

Of particular note is that MDS/AML may present with one or more of these features, either years before the onset of MDS/AML or in isolation with MDS/AML. MDS with germline *GATA2* variants is often associated with monosomy 7/del7q(-7) or trisomy 8, especially in children and young adults [205,208]. A study of 426 pediatric MDS cases identified germline *GATA2* variants in 37% of patients with primary MDS with 7 and 16% of MDS cases with trisomy 8 [209]. In contrast, no germline *GATA2* variants were found in treatment-related MDS. There have been over 480 individuals identified carrying a pathogenic or likely pathogenic germline GATA2 variant with symptoms of G2DS, with 240 of these confirmed to be familial and 24 de novo [57]. For those that develop myeloid malignancy (75% of all carriers with G2DS disease symptoms), the median age of onset is 17 years (range 0–78 years) and myelodysplastic syndrome is the first diagnosis in 75% of these cases with acute myeloid leukemia in a further 9% [57].

## 5. IBMFS

Inherited bone marrow failure syndrome (IBMFS) is an inherited disease associated with decreased bone marrow cell production [210,211,212]. It is associated with a specific clinical phenotype and variable risk of developing MDS or AML. Traditionally, the distinction has been made based on the presence or absence of classical physical manifestations [213] such as abnormal nails, reticulate pigmentation of the skin, and oral leukoplakia in congenital dyskeratosis. Fanconi anemia (FA) [214,215,216], Diamond-Blackfan anemia (DBA) [217,218,219], dyskeratosis congenita (DC) [220,221,222], telomere biology disorders (TBDs) [223], and Schwachman-Diamond syndrome (SDS) [224] are well-known predisposing factors for MDS/AML and exhibit characteristic physical symptoms and signs.

FA is an X-linked or autosomal recessive disorder characterized by genomic instability, hypersensitivity to DNA cross-linking agents, bone marrow failure, and predisposition to hematologic malignancies and solid tumors [210,211,212]. Hematologic abnormalities vary and include cytopenia, erythrocytosis, hypocellular bone marrow with mild dysplasia, and bone marrow failure with an increased risk of MDS or AML. The incidence of leukemia is even higher in the *FANCD1*/*BRCA2* subtype of FA, with most cases occurring at less than 5 years of age [225]. This clinically and genetically diverse syndrome is caused by germline mutations in any of at least 23 FA genes (*FANCA*-*FANCW*) that function cooperatively in DNA repair. The risk of progression to MDS or AML is very high (cumulative incidence of AML at age 50 years is 10% and MDS at age 50 years is 40%) [226]. Unlike other MDSs that are cured by HSCT, these patients have higher post-transplant morbidity and a higher risk of solid tumors compared to non-transplant patients.

DBA usually presents in infancy with macrocytic anemia and reticulocytopenia. Bone marrow histology usually shows aplasia of erythrocytes in normocytic bone marrow. Major causes of morbidity and mortality are associated with side effects of treatment and a long-term risk of malignancy [217,218,219]. X-linked variants in *GATA1*, which encodes a transcription factor important for erythropoiesis, are also a cause of DBA [227]. Disease mechanisms include p53-mediated apoptosis induced by ribosomal stress, increased cell death due to excess free heme with delayed globin production, increased autophagy, and translational changes in selective erythroid-specific transcripts such as *GATA1* [228].

DC/TBDs encompass genetically heterogeneous disorders associated with impaired telomere maintenance [220,221,222,223]. They are often associated with hematologic complications such as bone marrow failure, MDS, and AML. The cumulative incidence of MDS in DC/TBDs is estimated to be 2% by age 50 [229]. DC/TBD is associated with many non-hematologic complications, particularly pulmonary fibrosis, liver function abnormalities, and vascular abnormalities. Screening for TBD involves assessing the telomere length of lymphocytes, and further genetic testing for specific gene mutations is diagnostically useful because telomere shortening can also be seen in other diseases [230]. Telomeres shorten as the DNA replication cycle progresses. A critical shortening of telomere length leads to senescence and cell death [231].

SDS is characterized by pancreatic exocrine dysfunction and other physical findings. The most common nonhematologic abnormality is neurologic decompensation, which may be mild or severe, transient or persistent [224]. Other hematologic complications include bone marrow failure, MDS, and AML. In a French cohort of 102 SDS patients, the cumulative incidence of MDS/AML was 18.8% at age 20 and 36.1% at age 30 [232]. SDS is most often caused by an autosomal recessive mutation in the eponymous *SBDS* gene, resulting in low levels of SDS protein. SDS is involved in the binding of the large and small ribosomal subunits and functions as an elongation factor-like cofactor that removes the anti-binding factor eukaryotic initiation factor 6 (eIF6) from the large subunit [233]. SDS is also involved in the stabilization of mitotic spindles. The spectrum of *SBDS* variants, including missense, splice site, nonsense, frameshift, and partial or total gene deletions, has been confirmed. AML has been reported in patients with variants in the autosomal recessive gene in DnaJ Heat Shock Protein Family Member C21 genes (*DNAJC21*) and in those with various clinical features of SDS [234].

## 6. Infant Leukemia with a Germline Predisposition

Some infant leukemias with a germline predisposition have been reported and elucidated, although the section on them is not added in the 2016 revision of the WHO classification. Pediatric cancers typically harbor relatively few somatic mutations and frequently demonstrate developmentally immature phenotypes, suggesting a contribution of germline variation that might result in aberrant tissue development [235]. MLL rearrangements are observed in approximately 50–80% of infant ALL cases and 34–50% of infant AML cases [236]. There is evidence from multiple in vitro systems that the presence of a MLL rearrangement is insufficient by itself to drive leukemogenesis [237,238,239], suggesting that additional factors are required in the presence (and absence) of MLL rearrangements to drive leukemogenesis.

KMT2 protein is an epigenetic modifier, and each histone modification is associated with regulatory elements and mediates specific functions, enabling complex control over gene transcription [240]. KMT2C and KMT2D play an essential role in mediating monomethylation at histone 3 and lysine 4, primarily at enhancers [241]. Germline or somatic variations in a family of KMT2 lysine methyltransferases have been associated with a variety of congenital disorders and cancers. In mammals, somatic mutations of *KMT2C* and *KMT2D* are associated with various malignancies [242], with clear evidence for tumor suppressor roles [243,244]. Notably, KMT2A-fusions are prevalent in 70% of infant leukemias but fail to phenocopy short latency leukemogenesis in mammalian models, suggesting additional factors are necessary for transformation [245]. Heterozygous germline missense variants in KMT2C are more common in infant leukemia compared to healthy controls [246]. The loss of KMT2C in mice leads to aberrant myelopoiesis, causing myeloid infiltration into lymphoid organs; however, the loss of KMT2C alone is insufficient to drive leukemia [247]. Somatic cell drivers such as KMT2A fusions added to germline KMT2C mutations may more readily transform hematopoietic progenitor cells.

## 7. Conclusions and Perspectives

As discussed above, the genetic and phenotypic background of HHMS has been rapidly elucidated over the past decade, and the disease is now diverse. Most HHMS-related genes have clearly defined functions that contribute to hematopoietic regulation. However, the precise nature of this association requires further investigation. Advances in HHMS practice have been made possible by the introduction of next-generation sequencing (NGS) technology in germline and somatic gene testing. These tests now often have overlapping gene lists and have gained international recognition, especially for the diagnosis and management of myeloid malignancy. The association between germline genes predisposing to solid tumors and hematologic tumors is also becoming clearer. For example, variants in breast cancer gene type 1/2 (*BRCA1*/*2*), partner and localizer of BRCA2 (*PALB2*), and *TP53* occur in primary or treatment-related hematological malignancies, including AML, ALL, and MDS, narrowing the apparent distinction between solid tumors and hematologic tumor predisposition [248,249,250]. Future development of a hematologic tumor testing panel that is also useful in detecting refractory cytopenia and the risk of relapse refractoriness after leukemia-directed therapy is warranted. Extensive sequencing technologies, such as whole exome sequencing (WES), allow for the investigation of new candidate genetic abnormalities, including germline gene variants, at once, and are expected to be utilized more than targeted NGS panels in the future [4,251]. There is a growing need for the expert consultation and clinical surveillance of patients with a germline predisposition to hematologic malignancies [252]. Troublingly, prognosis and disease progression are slow. Therefore, consultation and treatment strategies must be tailored to the individual patient. Low-penetrance variants along with the late onset of the disease in some cases may be responsible for the delay in hereditary susceptibility recognition and have led some experts to propose universal germline testing strategies [248,253]. Bone marrow stromal cells show the advantage of being a readily available material from routine bone marrow aspirations, which can be isolated by culture [254]. However, confirmation in nonhematopoietic tissue or in other family members is necessary to avoid the misinterpretation of variants involved in CH, somatic copy number variants, or somatic loss heterozygosity [255]. For this purpose, skin fibroblasts are considered the gold standard, despite the requirement of a skin biopsy and long-lasting cultures. Patients and family members with suspected HHMS should be advised of the indications for genetic testing, the limitations of genetic testing, and genetic counseling. This is because curative therapy influences the outcome of allogeneic HSCT, regardless of the phenotypic spectrum or clinical presentation of HHMS [256]. The outcome in these patients is often poor, making them candidates for allogeneic HSCT. Compatible blood stem cell donors should be carefully considered, and donors with known germline variants or unknown retention status should be avoided. There are reports of cases of leukemia after allogeneic transplantation from blood donors [69]. *DDX41*, *CEBPA*, *GATA2*, and others have been reported to be present in 1~2% of allogeneic post-transplant relapses [257] with a median time of recurrence of 5.2 years [258]. There are also reports of onset 10 years after transplantation [70]. Various guidelines for genetic testing for HHMS are currently being proposed by organizations such as the National Comprehensive Cancer Network [259] and the American Society of Clinical Oncology [260]. However, rapid advances in the elucidation of the biology of hematologic tumors and in the clinical care of patients with these diseases necessitate the development of more detailed clinical guidelines. Providing clear eligibility criteria for HHMS testing, including the full spectrum of HHMS-related mutations, would improve the diagnosis and care of patients with these syndromes. Currently, no specific treatment for HHMS exists, and patients are not adequately treated. The lifelong surveillance of patients and their families is recommended to monitor for treatment-related toxicity, disease recurrence, and the development of new symptoms or signs in unaffected individuals. There is an international need to develop a comprehensive foundation for determining evidence-based management, family counseling, the treatment of symptomatic individuals, and preemptive interventions.

## Figures and Tables

**Figure 1 ijms-25-00652-f001:**
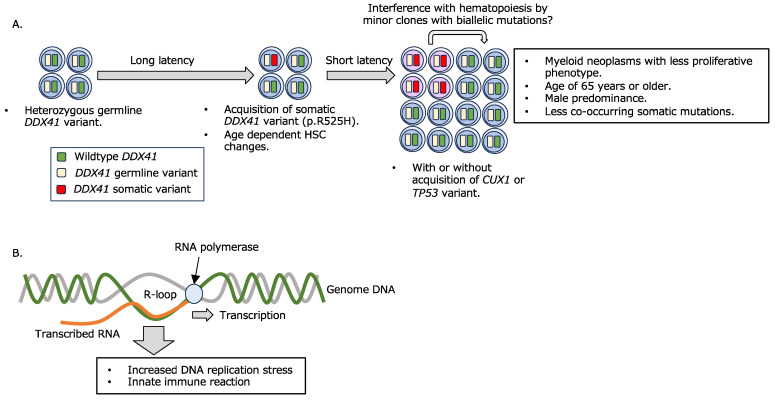
Involvement of *DDX41* variants in myeloid leukemogenesis. (**A**) Myeloid neoplasms arising from DDX41 variants: Hematopoietic cells carrying a heterozygous germline DDX41 variant (depicted as cells with blue nuclei) undergo the development of myeloid neoplasms following the acquisition of a somatic variant in the initially wild-type DDX41 after a prolonged latent period (illustrated as cells with light purple nuclei). The proportion of tumor cells tends to be low, and these cells may disrupt normal hematopoiesis, which is sustained by cells with only a germline variant. (**B**) Effects of R-loop accumulation on cellular function: R-loops form when transcribed RNA hybridizes with template DNA. The inappropriate accumulation of R-loops leads to DNA replication stress, impacting cellular function.

**Figure 2 ijms-25-00652-f002:**
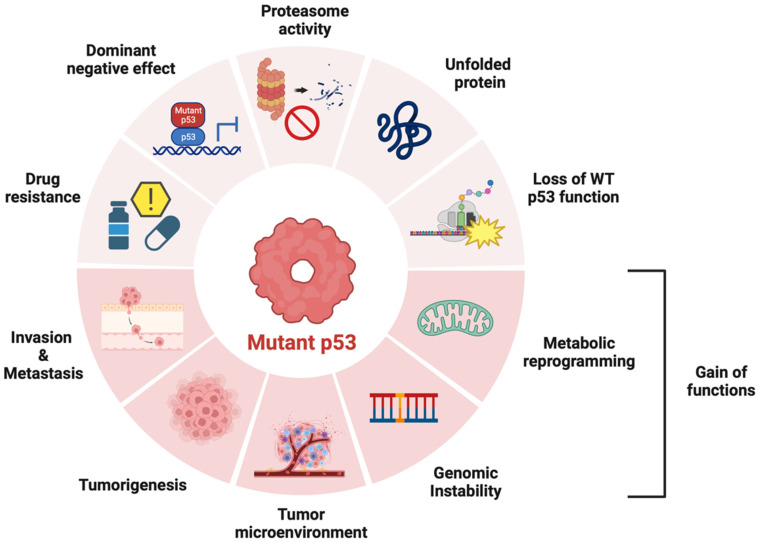
Role of p53 variants in cancer. p53 variants produce drug resistance, dominant negative effects on wild-type p53, proteasome repression, and LOF of wild-type p53. In cases of GOF, it promotes various cellular responses such as carcinogenesis, cancer cell proliferation, invasion, metastasis, tumor microenvironment establishment, genomic instability, and metabolic reprogramming.

**Figure 3 ijms-25-00652-f003:**
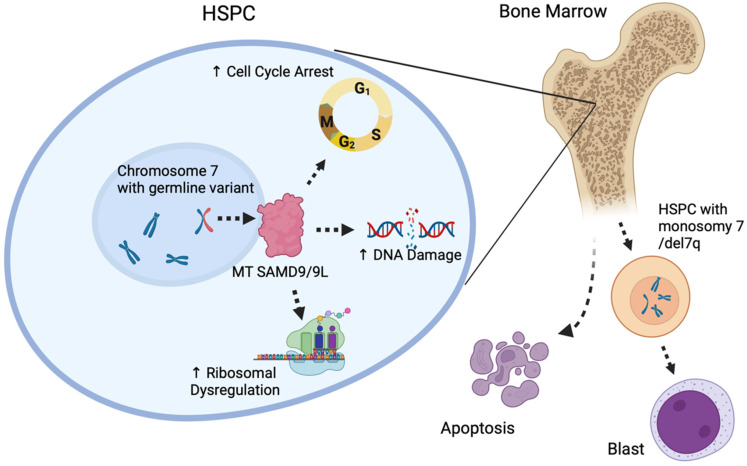
Role of *SAMD9* and *SAMD9L* in HSPC function. The *SAMD9* and *SAMD9L* genes regulate proteins involved in the cell cycle, DNA damage repair, and ribosome regulation. Mutant SAMD9 and SAMD9L proteins significantly enhance these functions, which cause decreased hematopoietic potential and apoptosis in the bone marrow, promoting monosomy 7/del 7 HSPC production. Hematopoietic stem and progenitor cell (HSPC), myelodysplastic syndrome (MDS), and mutant type (MT).

**Table 1 ijms-25-00652-t001:** Clinical characteristics, genetics, and prevalence of HHMS. Only the major genes discussed in this review are included in this table.

Gene	Chromosome Location	Disorder Name	Penetrance and Lifetime Risk of HM	Prevalence	Age of MN Onset, Years	Malignancy Types	Other Manifestations	Citations
*DDX41*	5q35.3	Familial MDS/AML with mutated *DDX41*	incomplete penetrance	Up to 13% of myeloid neoplasms have a genetic background, of which *DDX41* variants account for about 80% of cases.	Median age is 65 years, ranging from 44 to 88 years, which notably overlaps with the average age of sporadic myeloid malignancies.	MDS, AML, t-MN, solid tumors, especially colon and prostate cancer and melanoma, but not yet definitively linked	cytopenia, macrocytosis, autoimmune diseases	[13,14,18,19,20,21,22,23]
*TP53*	17p13.1	Li-Fraumeni syndrome (LFS)	lifetime risk of HM is about 6%	LFS affects all ethnicities and has an estimated incidence of 1:5000.	Nearly 100% of individuals develop cancer by the age of 70, with the median age of first cancer at 20 to 30 years.	MDS, AML, ALL, t-MN, lymphoma, MM, osteosarcoma, breast cancer, brain tumors, soft tissue sarcoma, adrenocortical carcinoma and other solid tumors	none	[7,24,25,26,27,28,29]
*CEBPA*	19q13.1	Familial AML with mutated *CEBPA*	>80% lifetime risk of AML	<20 families reported	Median age is 24.5 years, ranging from 2 to 50 years.	AML	none	[13,30,31,32,33,34]
*RUNX1*	21q22.12	Familial platelet disorder with propensity to myeloid malignancy	unknown	>250 families reported	Median age is 33 years, ranging from 6 to 76 years.	MDS, AML, ALL, other lymphoid malignancies	thrombocytopenia, platelet dysfunction, atopic and autoimmune disorders	[13,35,36,37,38,39,40,41]
*ANKRD26*	10p12.1	Thrombocytopenia 2	penetrance for thrombocytopenia is near complete, lifetime risk of HM is about 8%	Unknown	Median age is over 30 years, ranging from 20s to 70s years.	MDS, AML, CML, MPN, ALL, CLL, MM	thrombocytopenia, leukocytosis, erythrocytosis, mild bleeding tendency	[14,42,43,44]
*ETV6*	12p13.2	Thrombocytopenia 5	penetrance for thrombocytopenia is near complete	ALL is more frequent, especially in B-ALL (0.8% of unselected childhood B-cell ALL). The ratio of lymphoid versus myeloid malignancies is roughly 2:1.	Age ranges from 8 to 82 years and seem to occur at a younger age than usual but is not yet defined.	ALL, MDS, AML, CMML, MM, GI cancers	thrombocytopenia, macrocytosis, platelet dysfunction	[13,15,45,46,47,48,49,50]
*SAMD9*	7q21.2	MIRAGE Syndrome	unknown	8–17% of childhood onset MDS >110 individuals reported	Pediatric age, not yet defined.	MDS, AML, CMML	bone marrow failure, cytopenia, infections, growth restriction, adrenal hypoplasia, enteropathy, genital abnormalities	[13,51,52,53,54,55]
*SAMD9L*	7q21.2	Ataxia, Pancytopenia Syndrome					systemic autoinflammatory disease, bone marrow failure, ataxia	
*GATA2*	3q21.3	GATA2 deficiency syndrome	incomplete penetrance	>480 individuals reported, with 240 of these confirmed to be familial and 24 de novo	Median age is 17 years, ranging from 0 to 78 years.	MDS, AML, CMML, ALL	immunodeficiency, bone marrow failure, monocytopenia, lymphopenia, neutropenia, other cytopenia, infections, lymphedema, congenital deafness, pulmonary alveolar proteinosis, venous and arterial thrombosis	[13,37,56,57,58]

ALL, acute lymphoblastic leukemia; CML, chronic myeloid leukemia; CMML, chronic myelomonocytic leukemia; CLL, chronic lymphocytic leukemia; t-MN, therapy-related myeloid neoplasms; MM, multiple myeloma; MPN, myeloproliferative neoplasm; HM, hematological malignancies.

## Data Availability

Not applicable.

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
