# Peer review of "Germline Variants and Characteristic Features of Hereditary Hematological Malignancy Syndrome"

_ijms, 2024, doi:10.3390/ijms25010652_

Round 1
Reviewer 1 Report
Comments and Suggestions for Authors
The manuscript is very well written. I have no comments/ suggestions
Author Response
Reviewer 1:
The manuscript is very well written. I have no comments/ suggestions
Reply:
Thank you for the positive feedback. It is a great honor for us.

Reviewer 2 Report
Comments and Suggestions for Authors
The review article entitled: “Germline Variants and Characteristic Features of Hereditary Hematological Malignancy Syndrome” by Hironori Arai et al, presents the most prevalent germ line variants associated with hereditary hematopoietic malignancies (HHMs), which pose an increased lifetime risk for hematopoietic malignancies.
Although it is a very interesting topic and further studies are required to improve comprehension of HHM disorders enabling better clinical management, the present review has limited novelty compared to studies including large patient cohorts (https://doi.org/10.1182/bloodadvances.2023009742) or multi-institutional studies on the same subject (HHMs), interpreting results on the leukemogenic mechanisms underlying the germ line mutations (https://doi.org/10.1182/bloodadvances.2023010045). Also, this review follows the same structure and illustrates the impact of the same genetic loci as a current review article on Myelodysplastic Syndromes (clonal hematopoietic disorders), by Gener-Ricos et al, 2023 (DOI: 10.1097/PPO.0000000000000660).
Due to these reasons and lack of novelty publication is not proposed.
Comments on the Quality of English LanguageMinor editing of English language is required
Author Response
Reviewer 2:
The review article entitled: “Germline Variants and Characteristic Features of Hereditary Hematological Malignancy Syndrome” by Hironori Arai et al, presents the most prevalent germ line variants associated with hereditary hematopoietic malignancies (HHMs), which pose an increased lifetime risk for hematopoietic malignancies.
Although it is a very interesting topic and further studies are required to improve comprehension of HHM disorders enabling better clinical management, the present review has limited novelty compared to studies including large patient cohorts (https://doi.org/10.1182/bloodadvances.2023009742, Francesca Guijarro et al.) or multi-institutional studies on the same subject (HHMs), interpreting results on the leukemogenic mechanisms underlying the germline mutations (https://doi.org/10.1182/bloodadvances.2023010045). Also, this review follows the same structure and illustrates the impact of the same genetic loci as a current review article on Myelodysplastic Syndromes (clonal hematopoietic disorders), by Gener-Ricos et al, 2023 (DOI: 10.1097/PPO.0000000000000660).
Due to these reasons and lack of novelty publication is not proposed.
Reply:
Thank you very much for your careful peer review. We have added the latest findings on HHMS with references to the proposed literature. In addition, we enhanced Tables and Figures, which we believe are more novel than previously published papers.
Reviewer 3 Report
Comments and Suggestions for Authors
The review article submitted by Arai et al., titled "Germline Variants and Characteristic Features of Hereditary Hematological Malignancy Syndrome," is a well-documented piece. The authors delve into a crucial area that still harbors numerous unanswered questions, particularly as sequencing technologies advance. However, I have several concerns
Major Comments:
1) Mechanistic Details of Germline Variants: The authors appear to have overlooked significant mechanistic details concerning germline variants, such as the enhancer hijacking mechanism and its role in the development of AML progression. I recommend incorporating key studies into the main text:
a) PMID: 24301523: A study from WUSTL conducted targeted sequencing of infant leukemia cases, revealing an enrichment of non-synonymous germline variations in histone modifier genes like KMT2C, KMT2D, KDM6A, etc. The study also explores the mechanistic details of how these germline variations contribute to the disease. The authors demonstrate that KMT2CKO iPSCs fail to specify hemogenic endothelial cells in an in-vitro hematopoietic differentiation system (PMID: 34304711), providing a groundbreaking exploration of the role of germline mutation (loss of function) in blood development.
b) PMID: 34675061: Another study supporting similar conclusions should be discussed in the main text, emphasizing the mechanisms of germline variation in the context of AML progression. These studies are pioneering in unraveling the mechanisms of germline variation for a single gene in blood development, and the authors should carefully attend to these details to provide a comprehensive understanding of the field.
Minor Comments:
1) Prevalence and Population of HHMS:
- What is the prevalence of Hereditary Hematological Malignancy Syndrome (HHMS)? Additionally, provide information on the populations to which the reported HHMS cases mostly belong.
2) Age-specific Information:
- When discussing young patients, specify the age range. Some genes may have a long latency period, while others may lead to disease development at an early age. Categorize and discuss accordingly.
3) Inclusion of a Table:
- Consider adding a table in the main text that highlights relevant studies, including the ones described above (1a, 1b, and 2). Summarize their key findings to enhance reader interest and provide a quick reference for the intricacies of germline variation in blood development.
Addressing these major and minor comments will strengthen the review article by incorporating essential mechanistic details and contextualizing the findings within the broader landscape of germline mutations and AML progression.
Author Response
Reviewer 3:
The review article submitted by Arai et al., titled "Germline Variants and Characteristic Features of Hereditary Hematological Malignancy Syndrome," is a well-documented piece. The authors delve into a crucial area that still harbors numerous unanswered questions, particularly as sequencing technologies advance. However, I have several concerns
Major Comments:
1) Mechanistic Details of Germline Variants: The authors appear to have overlooked significant mechanistic details concerning germline variants, such as the enhancer hijacking mechanism and its role in the development of AML progression. I recommend incorporating key studies into the main text:
- a) PMID: 24301523:A study from WUSTL conducted targeted sequencing of infant leukemia cases, revealing an enrichment of non-synonymous germline variations in histone modifier genes like KMT2C, KMT2D, KDM6A, etc. The study also explores the mechanistic details of how these germline variations contribute to the disease. The authors demonstrate that KMT2CKO iPSCs fail to specify hemogenic endothelial cells in an in-vitro hematopoietic differentiation system (PMID: 34304711), providing a groundbreaking exploration of the role of germline mutation (loss of function) in blood development.
- b) PMID: 34675061:Another study supporting similar conclusions should be discussed in the main text, emphasizing the mechanisms of germline variation in the context of AML progression. These studies are pioneering in unraveling the mechanisms of germline variation for a single gene in blood development, and the authors should carefully attend to these details to provide a comprehensive understanding of the field.
Minor Comments:
1) Prevalence and Population of HHMS:
- What is the prevalence of Hereditary Hematological Malignancy Syndrome (HHMS)? Additionally, provide information on the populations to which the reported HHMS cases mostly belong.
2) Age-specific Information:
- When discussing young patients, specify the age range. Some genes may have a long latency period, while others may lead to disease development at an early age. Categorize and discuss accordingly.
We added details on the prevalence and age of each HHMS as well as epidemiological information in Table 1.
3) Inclusion of a Table:
- Consider adding a table in the main text that highlights relevant studies, including the ones described above (1a, 1b, and 2). Summarize their key findings to enhance reader interest and provide a quick reference for the intricacies of germline variation in blood development.
Addressing these major and minor comments will strengthen the review article by incorporating essential mechanistic details and contextualizing the findings within the broader landscape of germline mutations and AML progression
Reply:
Thank you very much for your careful peer review. I think the role and mechanism of germline mutation in blood developmental processes is very interesting. However, it is difficult to write a review article right away because it would be far from my field of expertise. In this article, I mainly focused on HHMS, which is discussed in the WHO 2016 edition, in light of the clinical picture and molecular biology of malignant tumors. We aim to make the content easily applicable to the daily practice of clinicians as well as researchers. There is a concern that the subject matter would become too large if we were to mention blood development. Since the deadline for revision has passed, we have decided to refrain from making any additions. We hope you will forgive us and give us your kind attention.
Round 2
Reviewer 2 Report
Comments and Suggestions for Authors
Authors have put an effort to improve their manuscript. However, there are still some major concerns as follows:
· The manuscript needs careful grammatical and syntactical editing.
· Table 1 should be reconstructed. The new Information included must be merged with the previous version of table 1 and edited accordingly.
· Authors should keep consistency in the way they present genes and genetic mutations involved in HHMS development. For example, a significant part is devoted to DDX41 and TP53 genes whereas, information on functioning and disease involvement for CEBPA, ATM and CHEK2 genes is hastily reported.
· Title of the following paragraphs: 2. (“Genes of Syndromes without Pre-Existing Disease or Organ Dysfunction”), 3. (“Genes of Syndromes Associated with Preexisting Platelet Disorders”) and 4. (“Genes of Syndromes Associated with Other Organ Dysfunction”) are weak and inadequately given. Paragraph titles should provide inclusive and accurate information.
· Subparagraph 4.3 reports an inherited disorder, the inherited bone marrow failure syndrome (IBMFS) and must be presented independently from subparagraphs 4.1 and 4.2, which present genetic alterations related to the subject of paragraph 4.
· Paragraph 5. (if it existed) is now missing.
· Conclusion section should be improved.
Comments on the Quality of English LanguageThe manuscript needs careful grammatical and syntactical editing.
Author Response
[Reply]
Thank you very much for taking the time to review our work.
We have corrected all the points you pointed out.
The tables have been combined into one.
We have shown the mechanism of CEBPA, ATM, CHEK2, etc.
Paragraph titles have been corrected and unified.
We enhanced the conclusion section.
Thank you very much for your time.
We would appreciate it if you could look over the areas you have indicated.

Reviewer 3 Report
Comments and Suggestions for Authors
It appears that the authors have not reached a consensus on addressing the reviewers' comments, and the response provided did not meet expectations. In the event that the submission deadline has elapsed, they have the option to request additional time for submission. Without incorporating and delving into the finer mechanistic details, it becomes challenging to identify novelty. Hence, I recommend that the authors revisit the manuscript, make necessary corrections, and thereby enhance the overall quality of the manuscript.
Author Response
[Reply]
Thank you very much for your very careful guidance.
We have corrected all the points you pointed out.
We have revised the pathogenesis of AML based on the information we received.
A paragraph on Infant leukemia with genetic predisposition was created and variants of histone modifiers in infant leukemia were mentioned.
In the paragraph on GATA2, the enhancer hijacking mechanism was explained in relation to AML.
HHMS prevalence, population, and age-specific information were added to the respective paragraphs, and a short summary of the main contents was included in table1.
I read with great interest the article you referred to PMID: 34675061 and was impressed by the fact that RUNX1 plays a very important role in hematopoiesis. However, since our review is focused on hematopoietic tumors caused by germline mutations, it was difficult to incorporate the content of PMID: 34675061.
We sincerely apologize for the inconvenience, but we would appreciate it if you could take the time to review our work again.

Round 3
Reviewer 3 Report
Comments and Suggestions for Authors
Authors incorporated all the suggested modifications. MS now improved significantly. I recommend this article for publication.